# Seeing Video Through Optical Scattering Media using Spatio-Temporal Diffusion Models

## Abstract

Optical scattering causes light rays to deviate from their trajectory, posing challenges for imaging through scattering media such as fog and biological tissues. Although diffusion models have been extensively studied for various inverse problems in recent years, its extension to video recovery, especially through highly scattering media, has been an open problem due to the lack of a closed-form forward model and the difficulty of exploiting the spatio-temporal correlation. To address this, here we present a novel inverse scattering solver using a video diffusion model. In particular, by deriving a closed-form forward model from the shower-curtain effect in a dynamic scattering medium, we develop a video diffusion posterior sampling scheme using a diffusion model with temporal attention that maximally exploits the statistical correlation between a series of frames and a series of scattered signals. Unlike previous end-to-end approaches only relied on spatial correlation between a scene and a scattered signal at specific settings, the adaptability of the proposed method is highly extendable to various types of scenes, various thicknesses of scattering media, and varying distances between a target scene and a medium. In particular, the use of temporal correlation is shown to be critical to faithfully retrieve high-frequency components which are often missed by inverse operations only in spatial domain. Simulation and real experimental results using various video datasets and real optical setup verify the effectiveness of the proposed method. To the best of our knowledge, this is the first video diffusion model to jointly utilize the correlations in both spatial and temporal domains in solving the inverse scattering problem. Code is available at https://github.com/video-through-scattering2023/video-through-scattering.

## 1 Introduction

Optical scattering is the phenomenon where light rays deviate from their original trajectory due to interactions with particles or irregularities in an inhomogeneous medium. This phenomenon arises in many real-world applications such as de-hazing (Berman et al., 2016; Satat et al., 2018), underwater imaging (Akkaynak & Treibitz, 2019), compressed sensing (Antipa et al., 2018), and fluorescent imaging (Alterman et al., 2021).

Recently, considerable success has been realized in the field of wavefront shaping, based on the deterministic measurement and manipulation of scattered optical waves through a static scattering medium (Mosk et al. (2012)). Inspired by the earlier works on deterministic approaches, several computational strategies have been proposed to use the statistical correlation (i.e. ensemble-averaged property) in scattered optical waves (Feng et al., 1988). For instance, the short-range correlation of scattered waves (i.e. memory effect) results in the translation invariance for the input and output planes positioned far from scattering media (Freund et al., 1988). Therefore, the spatial auto-correlation function of the object becomes identical to that of the scattered pattern within the memory effect range, resulting in the formulation of the inverse scattering problem into a phase-retrieval problem (Bertolotti et al., 2012; Katz et al., 2014). Furthermore, transmission matrix approach (Popoff et al., 2011), guidestar-assisted imaging (Xu et al., 2011; Horstmeyer et al., 2015), guidestar-free imaging (Yeminy & Katz, 2021; Feng et al., 2023), dynamic scatter imaging correlography (Edrei & Scarcelli, 2016), and end-to-end deep learning approaches (Li et al., 2019; Shi

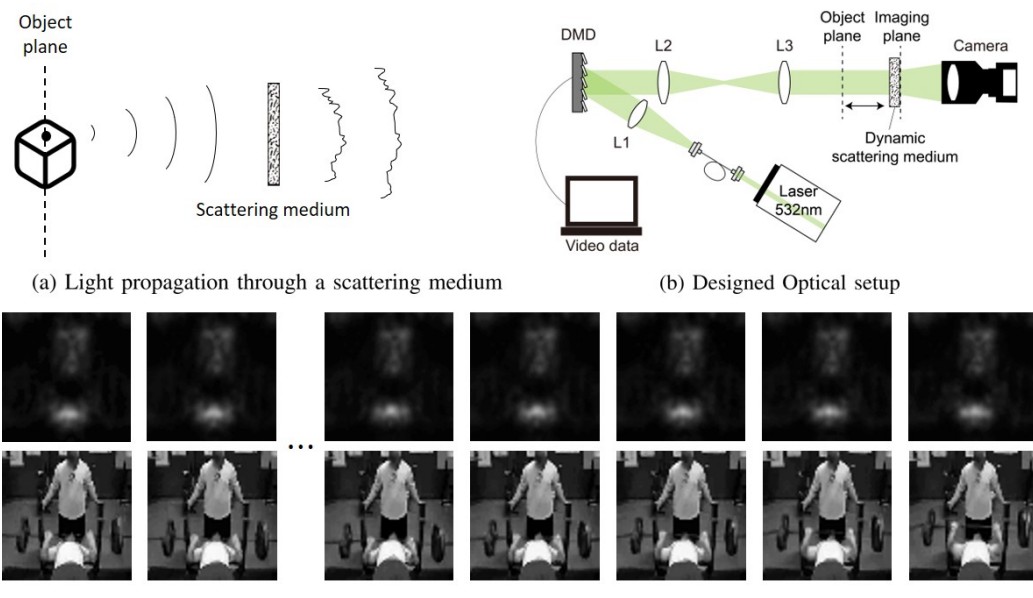

Figure 1: Solving the video inverse problem through a dynamic scattering medium. (a) Illustration of light propagation through a scattering medium, (b) designed optical setup, and (c) our reconstruction results (bottom) from the measurements (top) are shown.

et al., 2022) have also shown promising results, all of which address the imaging through scattering problem.

While successful in many cases, most of the previous approaches cannot faithfully recover high-frequency information beyond the range of memory effect (i.e. their reconstruction capability is limited to simple patterns, such as a single digit and letter) and also lack of adaptability to various types of scenes, various thicknesses of scattering media, and varying distances between a target scene and a medium. Also, considering the time scale of temporal decorrelation of scattering media in nature (e.g. shorter than 1 ms for mm-thick mouse skin slab), such approaches have limited capability in real-world applications (Jang et al., 2015). In this work, we aim to address those challenges using spatio-temporal diffusion models.

Specifically, diffusion models (Ho et al., 2020; Song et al., 2020; Nichol & Dhariwal, 2021) learn the prior distribution of the data $x$ by learning the score function, which is the gradient to the log density $\nabla_x \log p(x)$. Thanks to their superior generative capabilities and ease of integrating into iterative reconstruction, diffusion models have recently emerged as a trending class of inverse problem solvers (Choi et al., 2021; Song et al., 2020; Chung et al., 2022c;b; 2023; Song et al., 2022). While some of these prior methods require complex computations (e.g. singular value decomposition, transpose, pseudo-inverse of the forward operator), the recent proposal of diffusion posterior sampling (DPS) (Chung et al., 2022a) presents a simple yet general framework for solving inverse problems without the complex computations.

To incorporate the DPS into video reconstruction through a scattering medium, the forward operator should be derived into a closed form. In general, the physics of imaging through a scattering medium is almost impossible to express in closed form due to its randomness, which produces a random interference effect known as a speckle pattern. Unlike previous approaches, we therefore construct an optical setup to directly image the output surface of a scattering medium that allows us to formulate a scattering process as a sequence of wave propagator and convolution operation with a Gaussian kernel. With this form, we can parameterize the object-to-medium distance and the thickness of a medium as a propagator distance and a kernel width. Then, we develop a video diffusion model architecture that uses the temporal attention mechanism to exploit the temporal correlations of time-sequential frames. With the combined constraints of the data prior and the forward model in a spatio-temporal domain, we can adaptively deal with inverse scattering problems in different scene types and different imaging configurations with a single trained model.

Through extensive numerical and real experiments on various video datasets, we establish that our approach exhibits superior reconstruction quality compared to previous methods. In particular, we demonstrate that the approximation error of the 2D diffusion-based prior inverse problem solver can easily lead to temporally incoherent solutions, whereas the proposed method effectively mitigates this issue. Notably, our approach is the first to tackle the reconstruction of videos through a dynamic scattering medium.

## 2 RELATED WORKS

**Imaging Through Scatter.** In many imaging scenarios, recent advances demonstrate imaging through scattering based on their working thickness and scattering strength. As thick and weak scattering exponentially decays the magnitude of the ballistic light by Beer-Lambert Law, the real-world applications such as de-hazing (Berman et al., 2016; Satat et al., 2018) and underwater imaging (Akkaynak & Treibitz, 2019) reconstruct attenuation of ballistic light according to depth, color, or gamma information. As strong scattering disturbs light propagation in a random fashion resulting in a completely distorted input-output relationship, it has long been considered impossible to acquire well-resolved images through a scattering medium. Recently, considerable success has been realized in the field of wavefront shaping (Mosk et al., 2012; Katz et al., 2014), transmission matrix approach (Popoff et al., 2011), and guidestar-assisted imaging (Xu et al., 2011; Horstmeyer et al., 2015) through thick and strong scattering. In thin and strong scattering, the scattered waves result in the translation invariance(i.e. memory effect). Thus, guidestar-free imaging (Yeminy & Katz, 2021; Feng et al., 2023), dynamic scatter imaging correlography (Edrei & Scarcelli, 2016), and end-to-end deep learning approaches (Li et al., 2019; Shi et al., 2022) based on the imaging through scattering problem boil down to the traditional phase retrieval problem in far-field configuration.

Notably, there have been a few attempts to take account of imaging configurations and the effect of finite thickness of scattering media based on thorough considerations on scattering models and the range of correlation effects including non far-field configuration and thicker scattering medium using a single scattering model (Bar et al., 2021) and local support to enhance the effect of local correlation (Alterman et al., 2021).

Unlike previous approaches, our model is the first to combine a wave propagation model and a convolution with a blur kernel in the context of seeing through dynamic scattering media. With these modifications, one may deal with the problem of an object positioned at an arbitrary distance from the scattering medium and the medium's thickness. There is potential for its application in situations where rapid speckle decorrelation occurs due to blood flow and biological activities in scattering media within living tissues.

**Video Diffusion Models.** Diffusion models (Ho et al., 2020) attempt to model the data distribution $p_{\text{data}}(\boldsymbol{x})$ by the Markovian forward conditional densities

$$p(\boldsymbol{x}_t|\boldsymbol{x}_{t-1}) = \mathcal{N}(\boldsymbol{x}_t|\sqrt{\beta_t}\boldsymbol{x}_{t-1}, (1-\beta_t)I), \quad p(\boldsymbol{x}_t|\boldsymbol{x}_0) = \mathcal{N}(\boldsymbol{x}_t|\sqrt{\bar{\alpha}_t}\boldsymbol{x}_0, (1-\bar{\alpha}_t)I). \quad (1)$$

Here, the noise schedule $\beta_t$ is an increasing sequence of $t$, with $\bar{\alpha}_t := \prod_{i=1}^{t} \alpha_t$, $\alpha_t := 1 - \beta_t$. Training of diffusion models amounts to training a multi-noise level residual denoiser:

$$\min_{\theta} \mathbb{E}_{\boldsymbol{x}_t \sim q(\boldsymbol{x}_t|\boldsymbol{x}_0), \boldsymbol{x}_0 \sim p_{\text{data}}(\boldsymbol{x}_0), \boldsymbol{\epsilon} \sim \mathcal{N}(0,\boldsymbol{I})} \left[ \|\boldsymbol{\epsilon}_{\theta}^{(t)}(\boldsymbol{x}_t) - \boldsymbol{\epsilon}\|_2^2 \right]. \quad (2)$$

Video Diffusion Models (VDM) (Ho et al., 2022b) extend the 2D diffusion models to video data $\boldsymbol{X} = [\boldsymbol{x}^1, \ldots, \boldsymbol{x}^f]$ where $f$ denotes the number of the temporal frames. The main change comes from the training step Eq. (2), which now reads

$$\min_{\theta} \mathbb{E}_{\boldsymbol{X}_t \sim q(\boldsymbol{X}_t|\boldsymbol{X}_0), \boldsymbol{X}_0 \sim p_{\text{data}}(\boldsymbol{X}_0), \boldsymbol{\mathcal{E}} \sim \mathcal{N}(\boldsymbol{0},\boldsymbol{I})} \left[ \|\boldsymbol{\mathcal{E}}_{\theta}^{(t)}(\boldsymbol{X}_t) - \boldsymbol{\mathcal{E}}\|_2^2 \right], \quad (3)$$

where $\boldsymbol{\mathcal{E}}_{\theta}^{(t)}(\boldsymbol{X}_t)$ denotes the 3D diffusion model by introducing a particular type of 3D U-Net that contains temporal attention layers. Specifically, 3D U-Net (Ho et al., 2022b) replaces 2D convolution layers with 3D convolution layers and additional temporal attention layers with relative positional encoding. Recent high-fidelity video generation diffusion models (Ho et al., 2022a; Singer et al., 2022) also match the score function to the same neural network architecture due to its high generative quality of temporally coherent data (i.e., video data).

**Inverse Problem Solvers Using Diffusion Models.** In many physical situations, we encounter a scenario where we have measurements $\boldsymbol{y}$ derived from the original signal $\boldsymbol{x}$:

$$\boldsymbol{y} = \mathcal{A}(\boldsymbol{x}) + \boldsymbol{n}, \quad \boldsymbol{n} \in \mathbb{R}^n, \quad \boldsymbol{x} \in \mathbb{R}^d, \tag{4}$$

where $\mathcal{A}(\cdot)$ represents the general forward measurement process, and $\boldsymbol{n}$ represents the measurement noise. The inverse problems are in general ill-posed problems, as the mapping from $\boldsymbol{y}$ to $\boldsymbol{x}$ exhibits a one-to-many relationship and it is not straightforward to fully recover the original signal $\boldsymbol{x}$ from the measurements $\boldsymbol{y}$.

In diffusion-based inversion approaches, one line of works (Ho et al., 2022a; Saharia et al., 2022) attempt to directly learn the conditional distribution $p(\boldsymbol{x}|\boldsymbol{y})$ by learning the conditional score function $\nabla_{\boldsymbol{x}} \log p(\boldsymbol{x}|\boldsymbol{y})$. However, these methods require retraining the conditional score function whenever the conditions or physical settings change. Recent advances in conditional diffusion models (Kadkhodaie & Simoncelli, 2021; Song et al., 2020; Choi et al., 2021; Chung et al., 2022c;b) address this problem by incorporating projection-based measurement constraints while utilizing the unconditional score function $\nabla_{\boldsymbol{x}_t} \log p(\boldsymbol{x}_t)$. However, the projection-based approach is not straightforward and fails when the measurement process $\mathcal{A}$ is nonlinear or noisy (Chung et al., 2022a).

On the other hand, diffusion Posterior Sampling (DPS) (Chung et al., 2022a) is so general that it can explore noisy or nonlinear image inverse problems within conditional diffusion models that use the unconditional score function $\nabla_{\boldsymbol{x}_t} \log p(\boldsymbol{x}_t)$. Specifically, applying Bayes' rule, the conditional score function $\nabla_{\boldsymbol{x}_t} \log p(\boldsymbol{x}_t|\boldsymbol{y})$ unfolds as follows:

$$\nabla_{\boldsymbol{x}_t} \log p(\boldsymbol{x}_t|\boldsymbol{y}) = \nabla_{\boldsymbol{x}_t} \log p(\boldsymbol{x}_t) + \nabla_{\boldsymbol{x}_t} \log p(\boldsymbol{y}|\boldsymbol{x}_t). \tag{5}$$

where the first part is the unconditional score function. For the latter part, which is intractable to compute, DPS derives the approximated gradient of the log-likelihood using the posterior mean via Tweedie's formula:

$$\nabla_{\boldsymbol{x}_t} \log p(\boldsymbol{y}|\boldsymbol{x}_t) \simeq \nabla_{\boldsymbol{x}_t} \|\boldsymbol{y} - \mathcal{A}(\mathbb{E}[\boldsymbol{x}_0|\boldsymbol{x}_t])\|_2^2, \quad \mathbb{E}[\boldsymbol{x}_0|\boldsymbol{x}_t] = \frac{1}{\sqrt{\bar{\alpha}_t}} \left( \boldsymbol{x}_t - \sqrt{1-\bar{\alpha}_t} \boldsymbol{\epsilon}_\theta^{(t)}(\boldsymbol{x}_t) \right), \tag{6}$$

As (6) can be easily implemented for any degradation operator $\mathcal{A}$ using automatic differentiation, we can easily incorporate DPS for video diffusion-based recovery through a scattering medium as long as the forward operator $\mathcal{A}$ is differentiable.

# 3 SEE VIDEO THROUGH OPTICAL DIFFUSERS

## 3.1 CLOSED-FORM FORWARD OPERATOR FORMULATION

To achieve a closed-form description of the forward operation $\mathcal{A}$ for posterior sampling, we create an optical setup. This setup images the output surface of a dynamic scattering medium, minimizing perturbation from scattering. In this section, we detail how direct imaging of a scattering medium's output surface results in a closed-form forward model.

Existing studies about the memory effect from first principles (Feng et al., 1988) have modeled scattering as an infinitesimally thin phase mask. This assumption breaks down in the real world, limiting the field of view in the memory effect scheme.

We modeled the scattering medium as a dynamic thin scatter which does not have to be infinitesimally thin. If the scattering medium is not infinitesimally thin, the memory effect range is limited within $\theta_{max}^M$ as shown in Fig. 2 (bottom). Due to the memory effect range being limited, the beam projected to range $\theta > \theta_{max}^M$ results in random scattering fields. If the scattering property of the medium changes during the measurement duration (e.g. dynamic scatter), the effect of the random scattering field washed out as shown in Fig. 2, and the bell-shaped (Gaussian) average amplitude profile remains (Judkewitz et al., 2015). It summarizes the sequence of physical phenomena to arrive at the shower curtain effect on dynamic scatter.

In our imaging scheme, the wave propagation with distance $z = d_1$ of the given image is described as $\boldsymbol{x}_{z=d_1} = \left| \mathcal{F}^{-1} \left[ \mathcal{F}[\boldsymbol{x}] e^{-id_1\sqrt{k^2-k_x^2-k_y^2}} \right] \right|^2$ using the angular spectrum method (Goodman, 2005). After wave propagation with distance $d_1$, as described above, the propagated image $\boldsymbol{x}_{z=d_1}$ is blurred

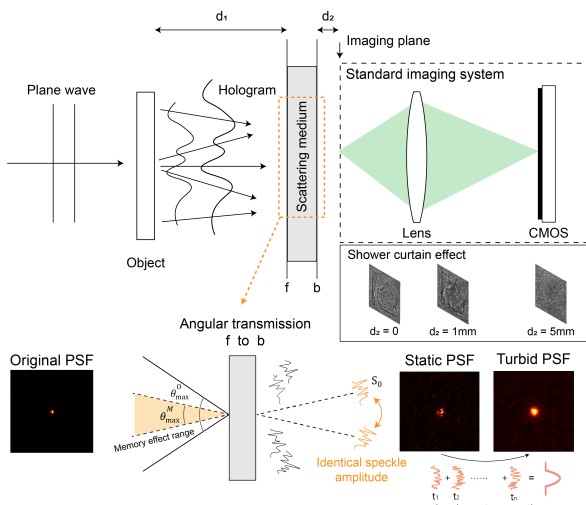

Figure 2: (Top) Forward propagation $z_0 = d_1$ of object light transmit through scattering medium and shower curtain effect appears when $d_2$ goes to 0. (bottom) Due to the memory effect, identical speckle summation in memory effect range for dynamic scattering medium results in Gaussian blur.

with a spatial kernel following the Gaussian profile. Therefore, the forward operator $\mathcal{A}$ of the optical setup focusing on the surface (i.e. $d_2 = 0$) of the dynamic scattering medium can be described as $\mathcal{A}(\boldsymbol{x}) = \left| \mathcal{F}^{-1} \left[ \mathcal{F}[\boldsymbol{x}] e^{-id_1 \sqrt{k^2 - k_x^2 - k_y^2}} \right] \right|^2 * h_\sigma$, where $\mathcal{F}[\cdot]$ represents 2-dimensional Fourier transform, $\mathcal{F}^{-1}[\cdot]$ represents 2-dimensional inverse Fourier transform, and $h_\sigma$ denotes the spatial Gaussian kernel with $\sigma$.

### 3.2 VIDEO DIFFUSION POSTERIOR SAMPLING

Given the closed-form expression of the forward operator, we aim to extend the DPS algorithm for video data. Since the optical forward operator $\mathcal{A}$ is defined as a 2-dimensional forward operator, one could use the DPS algorithm for each frame without considering the temporal correlation. However, for video inverse problems, if 2-dimensional posterior sampling (DPS) is applied frame-by-frame, each sampled frame can be seen as the desirable solution in image space, but it can easily lead to temporally incoherent solutions due to the approximation error of each sampled frame. This phenomenon can be understood from the geometric interpretation of DPS (Chung et al., 2022a). Specifically, a single denoising step in reverse diffusion sampling corresponds to the orthogonal projection of the data manifold, whereas the gradient step of DPS, i.e. $\zeta_i \nabla_{\boldsymbol{x}_t} \|\boldsymbol{y} - \mathcal{A}(\hat{\boldsymbol{x}}_0)\|_2^2$, takes a step tangent to the manifold (see Fig. 3(a)). Therefore, DPS tries to constrain the intermediate samples in the correct noisy data manifold while approaching the intersection between the clean manifold and the data consistency plane. Unfortunately, temporal correlation is not considered during the DPS, so there is no guarantee that resulting frame-by-frame solutions retain the desired temporal information.

To address this issue, we develop an extension to the video inverse problem, called Video Diffusion Posterior Sampling (VDPS). Specifically, instead of using 2D diffusion model as in Eq. (2), we employ 3D U-Net (Ho et al., 2022b) that contains temporal attention as a 3-D diffusion model $\mathcal{E}_\theta^{(t)}(\boldsymbol{X}_t)$. The 3D diffusion model is then trained by solving Eq. (3). Then, temporally coherent sampling can be implemented by ancestral sampling, which iteratively performs

$$\boldsymbol{X}_{t-1}' = \frac{1}{\sqrt{\alpha_t}} \left( \boldsymbol{X}_t - \frac{1 - \alpha_t}{\sqrt{1 - \bar{\alpha}_t}} \mathcal{E}_\theta^{(t)}(\boldsymbol{X}_t) \right) + \tilde{\beta}_t \mathcal{E} \tag{7}$$

$$\boldsymbol{X}_{t-1} = \boldsymbol{X}_{t-1}' - \zeta_i \nabla_{\boldsymbol{X}_t} \|\boldsymbol{Y} - \mathcal{A}(\hat{\boldsymbol{X}}_{0,t})\|_2^2 \tag{8}$$

where $\tilde{\beta}_t := \frac{1 - \bar{\alpha}_{t-1}}{1 - \bar{\alpha}_t} \beta_t$ and $\hat{\boldsymbol{X}}_{0,t}$ denotes the posterior mean from $\boldsymbol{X}_t$ by Tweedie's formula:

$$\hat{\boldsymbol{X}}_{0,t} = \frac{1}{\sqrt{\bar{\alpha}_t}} \left( \boldsymbol{X}_t - \sqrt{1 - \bar{\alpha}_t} \mathcal{E}_\theta^{(t)}(\boldsymbol{X}_t) \right), \tag{9}$$

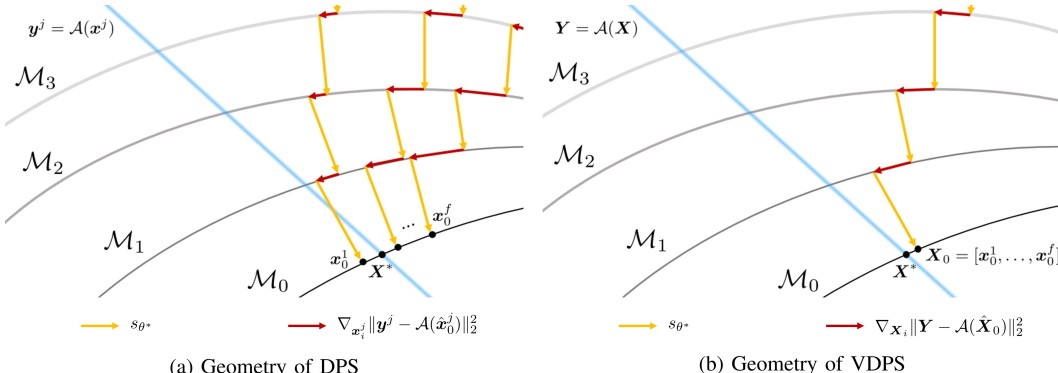

(a) Geometry of DPS

(b) Geometry of VDPS

Figure 3: Illustration of the geometry of the diffusion posterior sampling methods (a) DPS, (b) VDPS for video inverse problems.

Note that $\mathcal{E}_\theta^{(t)}$ is trained using the clean data sample from $p(\boldsymbol{X})$, so that the trained diffusion model learns the temporal correlation between the frames, so the denoising procedure using Eq. (9) becomes a projection to the clean manifold with temporal correlation. Therefore, the VDPS efficiently constraines the sampling path by supporting each individual frame to be temporally coherent (see Fig. 3(b)).

## 4 EXPERIMENTS

**Experimental Setup.** We have selected two datasets, UCF101 (Soomro et al., 2012) and VISEM-Tracking (Thambawita et al., 2023), to validate the performance of our method across diverse scenarios, including natural and biological scenes. We compared reconstruction performance on various physical settings, $(\sigma, d_1) = (0.5, 2.5), (1, 5), (2, 10)$, where $\sigma, d_1$ are defined in Section 3.1.

As baselines for comparison, we use the following methods: traditional convex optimization algorithm, Supervised, 2D DPS (Chung et al., 2022a), and VDPS (Ours). A traditional convex optimization algorithm was conducted by iterative optimizing the initial field $\boldsymbol{x}$ by minimizing the following loss $\||ASM(\boldsymbol{x})| - \boldsymbol{y}\|_2^2 + TV_{loss}(\boldsymbol{x})$, where $ASM$ is the angular spectrum method and $TV_{loss}$ is the total variation loss. The supervised method requires a paired dataset, which is not needed in our method and 2D DPS, suggesting another important advantages of diffusion models. We trained the supervised method by minimizing $L_2$ loss by taking measurements with physical setting $(\sigma, d_1) = (1, 5)$. For a fair comparison, every network architecture details of comparative methods and our method are identical except for convolution kernel shape (2D or 3D) and the existence of temporal attention layers. All comparative methods were trained in the same training implementations, including learning rate, training steps, and other relevant parameters. We validate our method works in real-world scattering measurements and shows the real measurements have strong correlations with in silico measurements. The validated results can be found in Appendix A. Also, to validate every method, the forward model was required by python implementation of closed-form forward physics which can be found in Appendix B. Further details, such as optical setup, datasets, hyperparameters, and network architectures can be found in Appendix C. Additional information on the experimental results in video format is described in Appendix D.

For a comprehensive quantitative comparison, we employ the standard video reconstruction performance metrics, including Peak Signal-to-Noise Ratio (PSNR) and Structural Similarity Index (SSIM) (Wang et al., 2004). Additionally, we utilized video-based metrics, such as Fréchet Video Distance (FVD) (Unterthiner et al., 2019), and image-based metrics, such as LPIPS (Zhang et al., 2018) calculated by averaging across frames, to evaluate perceptual qualities.

### 4.1 RESULTS

**UCF101** We first test our method on UCF101 (Soomro et al., 2012) containing natural scenes. We show the quantitative results of video reconstruction in Table 1. Also, representative results are illustrated in Fig. 4. We first observe that the proposed method shows highly accurate spatial

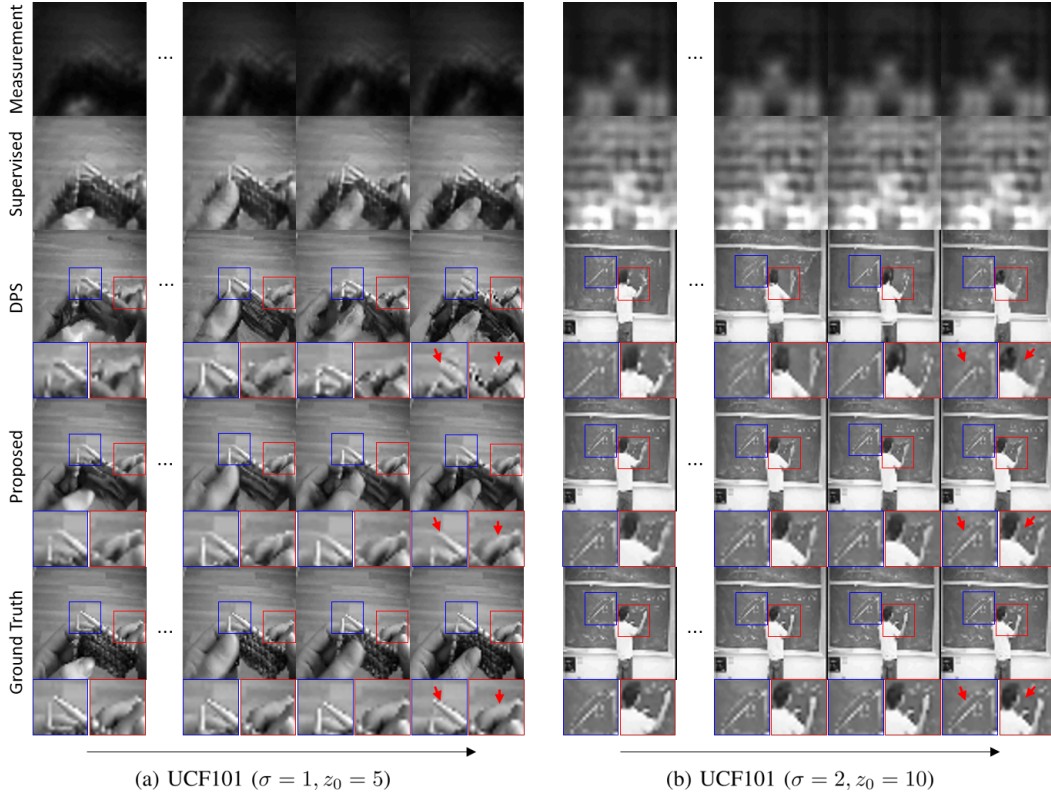

(a) UCF101 ($\sigma = 1, z_0 = 5$)        (b) UCF101 ($\sigma = 2, z_0 = 10$)

Figure 4: Results on solving video inverse problem for UCF101 with the following physical environments (a) $\sigma = 1, d_1 = 5$ and (b) $\sigma = 2, d_1 = 10$.

| | $\sigma = 1, d_1 = 5$ | | | | $\sigma = 2, d_1 = 10$ | | | | $\sigma = 0.5, d_1 = 2.5$ | | | |
|---|---|---|---|---|---|---|---|---|---|---|---|---|
| **Method** | PSNR $\uparrow$ | SSIM $\uparrow$ | LPIPS $\downarrow$ | FVD $\downarrow$ | PSNR $\uparrow$ | SSIM $\uparrow$ | LPIPS $\downarrow$ | FVD $\downarrow$ | PSNR $\uparrow$ | SSIM $\uparrow$ | LPIPS $\downarrow$ | FVD $\downarrow$ |
| VDPS (ours) | **26.218** | 0.841 | **0.039** | **227.06** | **22.712** | **0.712** | **0.083** | **332.02** | **30.168** | **0.913** | **0.019** | **164.48** |
| VDPS w/o TA | 24.121 | 0.760 | 0.078 | 557.90 | 22.631 | 0.697 | 0.087 | 613.62 | 26.205 | 0.820 | 0.060 | 488.62 |
| DPS | 23.044 | 0.725 | 0.081 | 704.17 | 19.806 | 0.563 | 0.159 | 781.39 | 27.734 | 0.864 | 0.029 | 513.51 |
| Supervised | **26.218** | **0.842** | 0.049 | 418.44 | 16.422 | 0.215 | 0.530 | 1514.2 | 14.844 | 0.190 | 0.299 | 1033.3 |
| Convex optimization | 9.6236 | 0.291 | 0.499 | 11920 | 10.744 | 0.247 | 0.590 | 2538.5 | 8.3335 | 0.343 | 0.4235 | 7879.8 |

Table 1: Quantitative evaluation (PSNR, SSIM, LPIPS, FVD) of solving video inverse problems on UCF101 test dataset. **Bold**: best, underline: second best.

reconstruction and temporal coherence, which consistently captures the high-frequency details along the given video measurement frames. Previous works mostly adopted supervised learning on UNet, thus we also compared the results of supervised learning. The supervised method mostly failed to reconstruct the high-frequency details and it led to producing blurry samples. Also, as shown in Table 1, the supervised method shows poor performance when the physical setting changes and the traditional convex optimization method mostly fails to reconstruct due to its high ill-posedness. DPS (Chung et al., 2022a) reconstructs realistic samples from given video measurement frames, however, the approximation error from each sampling trial easily leads to temporal inconsistency as shown in FVD metrics of Table 1. As shown in Fig. 4, we see that the proposed method not only produces temporally coherent solutions but also captures high-frequency details that were not captured from other comparative methods. Additionally, we observe that the proposed method is capable of generalization on various different video datasets. Note that we used the same score function which is trained on UCF101. As shown in Fig. 5, our method perfectly reconstructs high-frequency detailed samples from various degraded video measurements, while the score function never accessed a different dataset.

**VISEM** We test our method on VISEM-Tracking (Thambawita et al., 2023) containing biological scenes. We show the quantitative results of video reconstruction in Table 2. Also, representative results are illustrated in Fig. 6. As shown in Fig. 6, the proposed method captures high-frequency

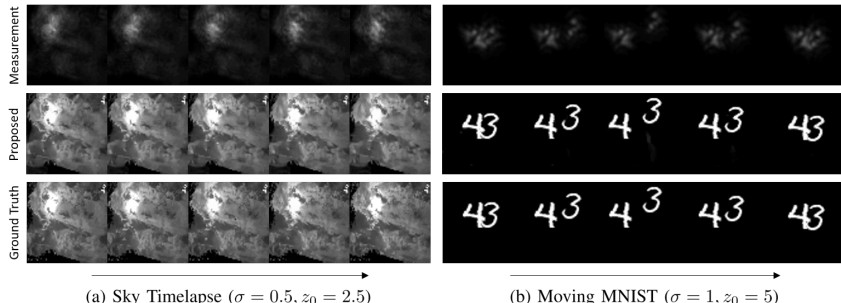

(a) Sky Timelapse ($\sigma = 0.5, z_0 = 2.5$)  (b) Moving MNIST ($\sigma = 1, z_0 = 5$)

Figure 5: Results on generalization test on different datasets which never seen to the score function $s_\theta$. (a) Sky Timelapse dataset (Xiong et al., 2018) and (b) Moving MNIST dataset (Srivastava et al., 2015).

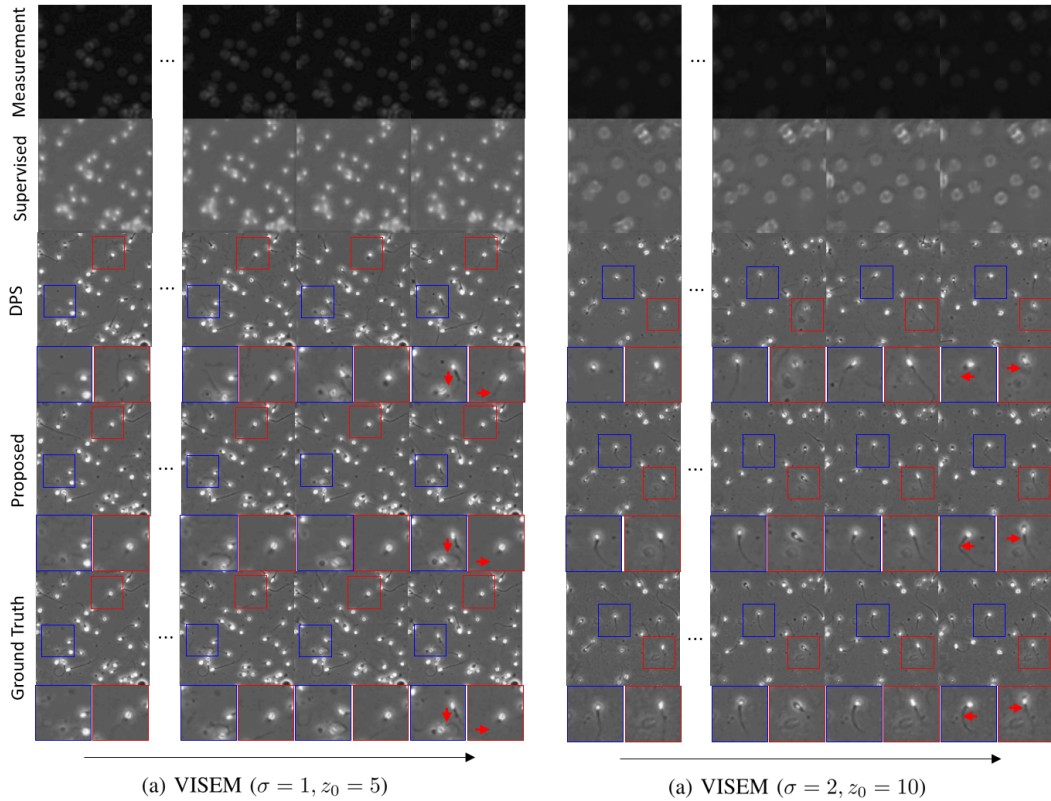

(a) VISEM ($\sigma = 1, z_0 = 5$)  (a) VISEM ($\sigma = 2, z_0 = 10$)

Figure 6: Results on solving video inverse problem for VISEM-Tracking with the following physical environments (a) $\sigma = 1, d_1 = 5$ and (b) $\sigma = 2, d_1 = 10$.

details of the tail of Spermatozoa, while the comparative methods fail to reconstruct the corresponding details. Again, the supervised method produced blurry samples which are worse than the UCF101 reconstruction. The reason is the overfitting problem due to the relatively small training set. On the other hand, diffusion-based reconstruction consistently reconstructs high-frequency details well even though the training set is relatively small. Also, the supervised method shows poor performance when the physical setting changes, and the traditional method fails to reconstruct as shown in Table 2. DPS reconstructs realistic samples from given video measurement frames, however, the approximation error from each sampling trial easily leads to temporal inconsistency as shown in FVD metrics of Table 2.

We additionally observe the temporal attention layers in 3D UNet are the key component for reconstructing video measurements. We manually removed temporal attention layers from 3D UNet to train the score function (dubbed as 'VDPS w/o TA' in Table 1 and 2). The remaining network archi-

| Method | $\sigma = 1, d_1 = 5$ | | | | $\sigma = 2, d_1 = 10$ | | | | $\sigma = 0.5, d_1 = 2.5$ | | | |
|---|---|---|---|---|---|---|---|---|---|---|---|---|
| | PSNR ↑ | SSIM ↑ | LPIPS ↓ | FVD ↓ | PSNR ↑ | SSIM ↑ | LPIPS ↓ | FVD ↓ | PSNR ↑ | SSIM ↑ | LPIPS ↓ | FVD ↓ |
| VDPS (ours) | **31.415** | **0.852** | **0.046** | **141.61** | **28.148** | **0.755** | **0.083** | **204.32** | **33.209** | **0.882** | **0.033** | **130.79** |
| VDPS w/o TA | 28.277 | 0.755 | 0.103 | 767.01 | 26.708 | 0.707 | 0.122 | 841.74 | 29.389 | 0.778 | 0.094 | 748.87 |
| DPS | 29.720 | 0.799 | 0.070 | 751.59 | 27.052 | 0.718 | 0.102 | 801.07 | 31.347 | 0.830 | 0.056 | 724.47 |
| Supervised | 26.217 | 0.755 | 0.257 | 1973.0 | 22.902 | 0.599 | 0.409 | 1663.1 | 24.858 | 0.712 | 0.310 | 1727.1 |
| Convex optimization | 9.8328 | 0.241 | 0.685 | 1984.5 | 10.159 | 0.247 | 0.793 | 2096.7 | 9.5659 | 0.229 | 0.600 | 1902.0 |

Table 2: Quantitative evaluation (PSNR, SSIM, LPIPS, FVD) of solving video inverse problems on VISEM-Tracking test dataset. **Bold**: best, underline: second best.

tectures and implementation details are identical to the original work. As shown in Table 1 and 2, we observe temporal attention layer plays a key role in improving video reconstruction performance.

## 5 CONCLUSION

Navigating the challenges of optical scattering, our work innovatively leveraged spatio-temporal correlations alongside the previous spatial approach to solve the inverse scattering problem. By integrating DPS with the video diffusion model and the closed-form forward physics, we've enhanced the exploitation of correlations across varied scenes and scattering conditions. This holistic approach, distinct from prior spatial-only methods, successfully captures high-frequency components, as validated by our tests on sperm cell video datasets. Our study pioneers the combined use of both spatial and temporal correlations in this domain.

### ETHICS STATEMENT

The ability to restore temporally coherent signals beyond windows or opaque objects can be exploited for privacy or personal data leakage. As described in Thambawita et al. (2023), the original study for the VISEM-Tracking dataset was approved by the Regional Committee for Medical and Health Research Ethics, South East, Norway (REK number: 2008/3957), and the original project was finished in December 2017, and all data was fully anonymized.

### REPRODUCIBILITY STATEMENT

A link to an anonymous downloadable source code is available with a provided anonymous GitHub page https://github.com/video-through-scattering2023/video-through-scattering. A complete proof of the closed-form forward physics is provided in Section 3.1 and its implementation code is provided in Appendix B. A complete description of the details of hyperparameters, training details, and compute resources used for each model are provided in Appendix C. A complete description of data processing steps is provided in Section 4.

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

## A  RESULTS IN REAL EXPERIMENTS

### A.1  OPTICAL SETUP DETAILS

Laser (Cobolt samba, 532nm) was spatially filtered by single-mode fiber and collimated with a lens ($f_{L1}$=150mm). We use a digital micromirror device (DMD) (Vialux V-7001, 13.68$\mu$m pixel size, 1024x768) to display amplitude video object. The object is conjugated to the object plane with 1x magnification by relaying optics ($f_{L2}$=100mm, $f_{L3}$=100mm). Departing from the object plane by a curtain distance, frontside of a turbid medium is placed. A digital camera was constructed by a CMOS sensor (BFS-U3-200S6M-C, 4.5$\mu$m pixel pitch) and a commercial imaging lens (Nikkon). The imaging plane was focused on the backside of the turbid medium. A turbid medium was realized by a scattering medium and DC motor stage. Displaying on a frame on the DMD was synchronized with the CMOS image acquisition. The overall illustration of the optical setup is shown in Fig. 7.

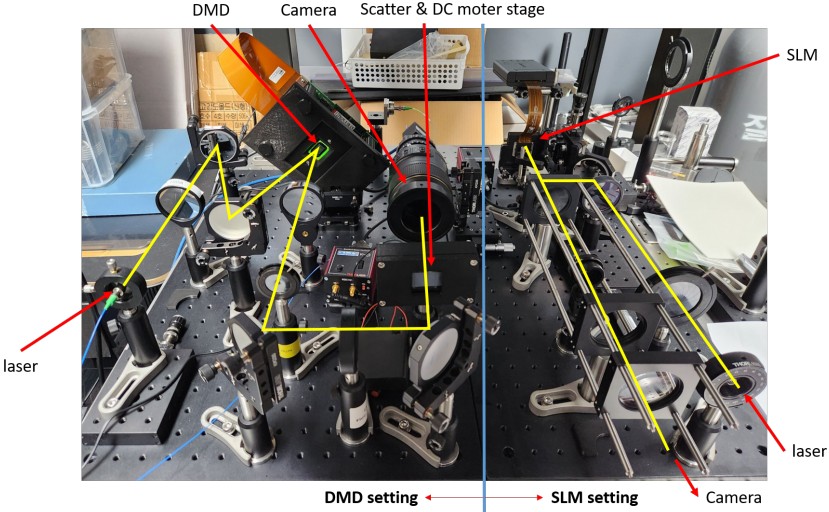

Figure 7: Experimental setup photograph. We create amplitude objects in the DMD setting and we create phase objects in the SLM setting. To show the both objects work on the forward physics, we reconstruct video from the DMD setting and demonstrate a strong correlation between real and in silico measurements in the SLM setting.

### A.2  PSF OF VARIOUS SCATTERING CONFIGURATION



Figure 8: Measured PSFs of various scattering configurations (a) with 1 scotch tape, (b)-(e) with 2 scotch tapes with various inter-layer distances.

We acquire the PSF of the dynamic scattering medium by focusing the point source through a lens and placing it right behind the scattering medium. Point response is then measured beyond the scattering medium. As shown in Fig. 8, the resulting PSF of the dynamic scatter with amplitude object was a 2d Gaussian function. It shows that in the real experiment setting, the forward model of dynamic scattering medium focused on the backside results in Gaussian blur as we described in Section 3.1.

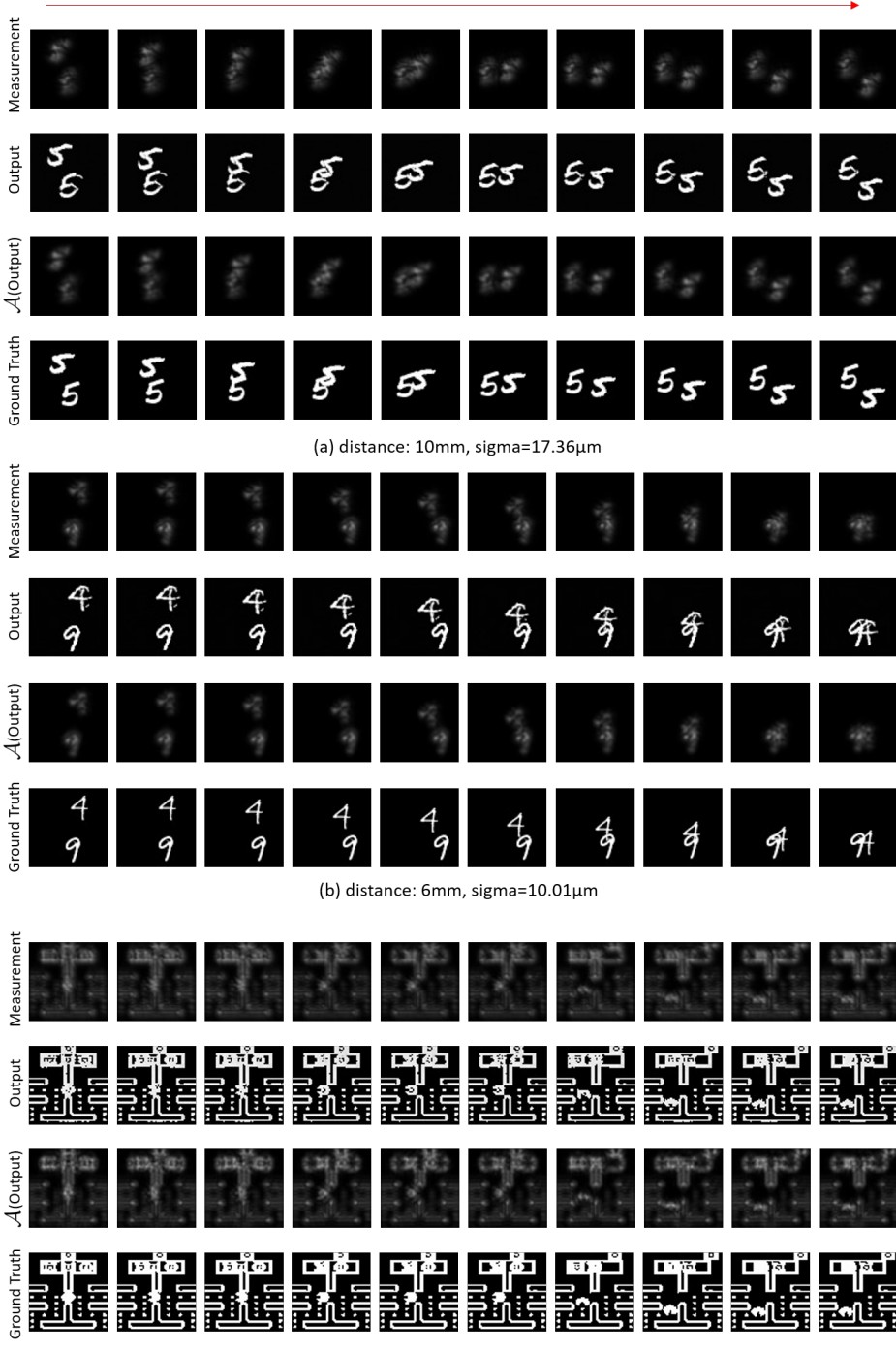

Figure 9: Results for the reconstruction of moving MNIST real measurements with the various settings in terms of propagation distance and the sigma of the Gaussian blur. (a) 10mm, $17.36\mu$m, (b) 6mm, $10.01\mu$m. And (c) the reconstruction of a Pac-Man video sourced from YouTube for 6mm, $17.36\mu$m. $\mathcal{A}$(output) represents a forwarded output.

Here, we provide a detailed experimental setting to acquire the PSFs. A point source with a size of 1 $\mu$m was focused in front of the scattering medium using a 0.2 NA plano-convex lens, and the imaging from the backside was captured using a 0.28 NA 10x objective lens. The scattering

medium used was 3M Scotch Magic Tape (previously used in the paper: Malone, Joseph D., et al. "DiffuserSpec: spectroscopy with Scotch tape." Optics Letters 48.2 (2023): 323-326). One or two layers of this tape were stacked, and a 50 $\mu$m thick Kapton tape was used as a spacer between the two layers to adjust the distance.

## A.3  RESULTS IN REAL EXPERIMENTS

From the real optical setting, we position the actual amplitude target at a distance 6/10mm from 2 different scattering mediums(1 scotch tape / 3 scotch tapes) and get the image in the real measurement.

Also, as shown in Fig. 9, our method is still effective while shot noise and camera dark noise are added. The result shows the tolerance of our method for model mismatches in real-world settings. Our method is effective in various settings in terms of propagation distance and the sigma of the Gaussian blur using a single pre-trained diffusion prior. Also, forwarded outputs show the same pattern as the measurements. The reason why the resulting PSF may not seem not perfectly Gaussian in Fig. 8 is that the lateral movement axis of the DC stage may not perfectly perpendicular to the incident light. The result shows that our method can cover the corresponding error to reconstruct the object. Additionally, we reconstruct more complex scenes from a Pac-Man video sourced from YouTube(https://youtu.be/jicSDEL2hZU). We crop the scene into 128×128 resolution, resize the scene into 64×64 resolution, and display the sequence of frames on the DMD. The acquisition process of the real measurement is identical to the moving MNIST dataset case. We reconstruct the real measurement using a diffusion prior trained from randomly sampled video sequences of the Pac-Man video that do not overlap with the test set. The result shows that the method can handle more complex scenes including a video sourced from YouTube.

## A.4  CORRELATION BETWEEN REAL MEASUREMENTS AND IN SILICO MEASUREMENTS OF PHASE OBJECT

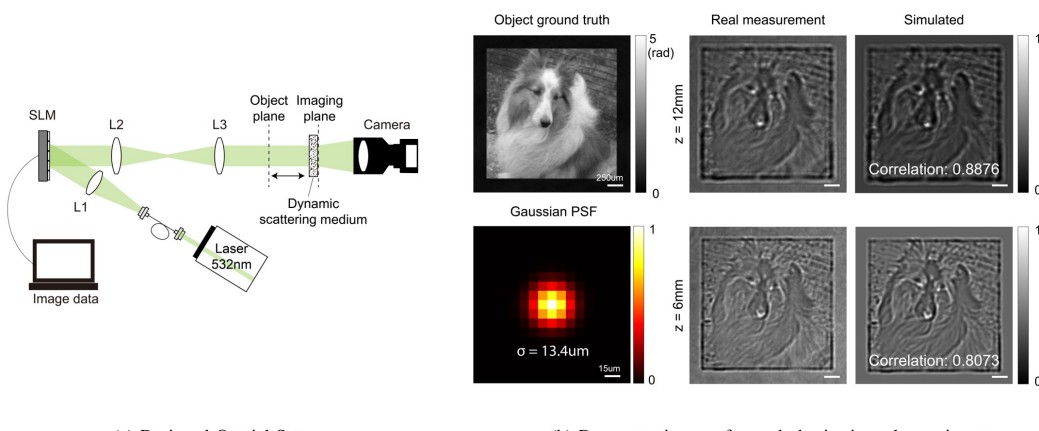

(a) Designed Optcial Setup

(b) Demonstration our forward physics in real experiment

Figure 10: Demonstration on our forward physics model in real experiment. (a) Designed optical setup with SLM, (b) Resulting PSF of the dynamic scattering medium was 2d Gaussian, and the correlations between real measurement and simulated output from our forward physics were strong.

We also demonstrate how well our forward model models the imaging of a complex phase object located at a distance $d$ behind a dynamic scattering medium. First, according to our forward model, we propagate a phase object by 6/12mm using the angular spectrum method. We take the intensity and convolve it with the Gaussian PSF of the scattering medium, resulting in the image in the simulated column. Then, we position the actual phase target at a distance of 6/12mm from the scattering medium and get the image in the real measurement as shown in Fig 10 (a). A two-layer stack of opaque Scotch tape, separated by 50 microns, was used as the scattering medium. To introduce turbidity, this medium was temporally vibrated along the lateral plane. The correlation

between the two images is calculated using the following formula, with a maximum value of 1:

$$r(A, B) = \frac{\sum\limits_{m}\sum\limits_{n}(A_{mn} - \bar{A})(B_{mn} - \bar{B})}{\sqrt{(\sum\limits_{m}\sum\limits_{n}(A_{mn} - \bar{A})^2)(\sum\limits_{m}\sum\limits_{n}(B_{mn} - \bar{B})^2)}} \tag{10}$$

where, $\bar{X}$ is the mean value of image $X$.

The resulting PSF of this turbid medium was a 2d Gaussian function with a standard deviation $13.4\mu$m. The 2D spatial correlations between the real measured intensity and the simulated intensity, as predicted by our forward model, exceeded 0.8 in both the 6mm and 12mm propagation cases as shown in Fig. 10.

### A.5 Sources for Model Miss Match

Several aspects were neglected from our forward model.

**Stochastic Noise**: Shot noise, camera dark noise, and other noises can be inserted into the measurement. However, in the case of shot noise(Poissonian), DPS (Chung et al., 2022a) has already shown that the posterior sampling method is effective in reconstructing noisy measurements including Poisson noise. Therefore, we neglect it from our model.

**Scattering Kernel**: Scattering kernel that describes the ensemble-averaged intensity response of a scattering medium (i.e. the intensity response on the output plane for a point source on the input plane) varies depending on the properties of a scattering medium, such as scatterers' shape and size, and refractive indices of scatterers and an embedding medium. Here, based on the assumption that the kernel function is smoothly and slowly decaying, we simply assumed it as a Gaussian function.

## B PYTHON IMPLEMENTATION OF CLOSED-FORM FORWARD PHYSICS

```python
input video_frame

## physical parameters
pix = 8e-6 # pixel size of DMD
lamb = 532e-9 # wavelength of laser
dist = 2.5 or 5 or 10 # forward propagation distance
sigma = 2 or 1 # Gaussian blur sigma
image_length = 128 or 64
FOV = 128 or 64
pad = 64 or 32

## define propagation
fx, fy = torch.meshgrid(torch.linspace(0, FOV + 2 * pad - 1, FOV + 2 *
                                       pad), torch.linspace(0, FOV + 2 *
                                       pad - 1, FOV + 2 * pad))

fx = (fx - np.fix((FOV + 2 * pad) / 2)) / ((FOV + 2 * pad) * pix)
fy = (fy - np.fix((FOV + 2 * pad) / 2)) / ((FOV + 2 * pad) * pix)

quad_pha = 1 / (lamb ** 2) - fx ** 2 - fy ** 2

prop = torch.exp(1j * 2 * torch.pi * dist * 1e-3 * torch.sqrt(quad_pha)).
                                       cuda()

## forward propagation
amp = video_frame
pha = torch.zeros((image_length,image_length))

field_full = amp * torch.exp(1j * pha)
field_pad = torch.nn.ReplicationPad2d(pad)(field_full)
field_fourier = torch.fft.fftshift(torch.fft.fft2(field_pad))

ASM = field_fourier * prop

diff_field = torch.fft.ifft2(torch.fft.ifftshift(ASM))
diff_field = diff_field[pad:-pad, pad:-pad]

inten = abs(diff_field) ** 2

## Shower-curtain effect
measurement = T.GaussianBlur(int(3*sigma)*2+1, sigma=sigma)(inten)

return measurement
```

## C  DETAILS AND HYPERPARAMETERS

### C.1  EXPERIMENTAL DETAILS

For UCF101, we divide a total of 13,320 videos into Train(9,437)/Test(3,783) splits based on official descriptions[1] of Train/Test video list 1 for action recognition on UCF101 data. The 3D diffusion model was taken from Ho et al. (2022b) and trained for UCF101 data from scratch using randomly selected 10 adjacent frames of the UCF101 training set for 1M steps. The 2D diffusion model was taken from Ho et al. (2022b) by eliminating temporal attention layers and modifying the 3D convolution kernels into 2D convolution kernels. The 2D diffusion models for UCF101 data was trained from scratch using randomly selected frames of UCF101 training set for 1M steps. We validate their performance using the UCF101 test set. To express the UCF101 dataset to amplitude object, we converted all videos to grayscale and normalized them within the range [0,1]. Additionally, we resized the UCF101 dataset to 64×64 pixels. VISEM-Tracking dataset divided 502 30 s video clips into 166 annotated videos and 336 unlabeled videos. Similarly, we divide a total of 502 videos into Train(166)/Test(336) splits based on official descriptions (Thambawita et al. (2023)) of annotated/unlabeled video sets. The 2D diffusion model for VISEM-Tracking data was trained from scratch using randomly selected frames of VISEM-Tracking training set for 1M steps. The 3D diffusion model for VISEM-Tracking data was trained from scratch using randomly selected 10 adjacent frames of VISEM-Tracking training set for 1M steps. We validate their performance using the VISEM-Tracking test set. To express the VISEM dataset to amplitude object, we converted all videos to grayscale and normalized them within the range [0,1]. Additionally, we downsampled by a factor of 2, randomly cropped to 128×128 pixels.

Here, we list the hyperparameters, training details, and compute resources used for each model. The following expression style was provided by Ho et al. (2022b). Every diffusion model and supervised method was trained with the following details.

### C.2  UCF101

| | |
|---|---|
| Base channels: 128 | Optimizer: Adam ($\beta_1 = 0.9$, $\beta_2 = 0.99$) |
| Channel multipliers: 1, 2, 3, 4 | Learning rate: 0.0001 |
| Blocks per resolution: 2 | Batch size: 8 |
| Attention resolutions: 8, 16, 32 | EMA: 0.995 |
| Attention head dimension: 32 | Dropout: 0.0 |
| Conditioning embedding dimension: 512 | Training hardware: 1 RTX 3090 GPU |
| Conditioning embedding MLP layers: 2 | Training steps: 1,000,000 |
| Diffusion noise schedule: cosine | Joint training independent images per video: 0 |
| Noise schedule log SNR range: [-20, 20] | Sampling timesteps: 1000 |
| Video resolution: 10x64x64 frameskip 1 | Sampling log-variance interpolation: $\gamma = 0.0$ |
| Weight decay: 0.0 | Prediction target: $\epsilon$ |

### C.3  VISEM-TRACKING

| | |
|---|---|
| Base channels: 128 | Conditioning embedding dimension: 512 |
| Channel multipliers: 1, 2, 3, 4 | Conditioning embedding MLP layers: 2 |
| Blocks per resolution: 2 | Diffusion noise schedule: cosine |
| Attention resolutions: 8, 16, 32 | Noise schedule log SNR range: [-20, 20] |
| Attention head dimension: 16 | Video resolution: 10x128x128 frameskip 1 |

---

[1] https://www.crcv.ucf.edu/data/UCF101.php

| | |
|---|---|
| Weight decay: 0.0 | Training hardware: 1 RTX 3090 GPU |
| Optimizer: Adam ($\beta_1 = 0.9$, $\beta_2 = 0.99$) | Training steps: 1,000,000 |
| Learning rate: 0.0001 | Joint training independent images per video: 0 |
| Batch size: 2 | Sampling timesteps: 1000 |
| EMA: 0.995 | Sampling log-variance interpolation: $\gamma = 0.0$ |
| Dropout: 0.0 | Prediction target: $\epsilon$ |

3D diffusion model was trained on the same neural network architecture from Ho et al. (2022b). We modify 3D convolution kernels to 2D convolution kernels and eliminate temporal attention layers to make the neural network for the 2D diffusion model and supervised method. The step sizes in (8) are defined as $\zeta_i = \sin(i\pi/1000)^2/||\boldsymbol{Y}||$. For the traditional convex optimization method, we optimize the initial field for 700 iterations using Adam optimizer with learning rate $10^{-3}$, and $TV$ loss weight is $3 \times 10^{-5}$.

## D   ADDITIONAL EXPERIMENTAL VIDEO RESULTS

We recommend seeing the experimental results in video format (e.g. gif format) to observe the temporal correlation of reconstructed output compared with the comparative methods. We provide 5 randomly (non-cherry-picked) reconstructed results for supervised, 2D DPS, and VDPS (ours) in Supplementary materials, please check to validate the superior performance of our method.

