# OpenReview forum: "Seeing Video Through Optical Scattering Media using Spatio-Temporal Diffusion Models"
_ICLR.cc/2024/Conference — Submitted to ICLR 2024_

### Official Review · Reviewer_dFYB · 2023-10-24

**Soundness:** 3 good
**Presentation:** 3 good
**Contribution:** 2 fair
**Rating:** 6
**Confidence:** 3

**Summary:**

This paper considers the problem of seeing dynamic scenes through scattering media. The authors propose a 3D convolution architecture that can take into account temporal correlation for the task of de-scattering the video sequence. Using a diffusion model prior regularizes the solution space of the inverse problem.

**Strengths:**

The paper tried to tackle a difficult problem using state-of-the-art generative AI approaches. They show that taking into account temporal correlations helps with the reconstruction of dynamic scenes undergoing scattering. They validate the approach as compared to a traditional TV approach and a 2D-based deep learning approach and show that using deep learning in video space improves the reconstruction quality.

**Weaknesses:**

The key issue and the reason I chose a rating of 2 for the presentation is the lack of contextualization for this work. While the authors do a good job comparing to some baselines like a TV method and a 2D approach the problem of seeing through scattering media has a long history, which this paper largely ignores. This problem arises in the context of de-hazing and underwater imaging (see e.g. Akkaynak et al and Berman et al below ). I also point out that the inverse problem is very similar to the DiffuserCam proposed by Antipa et al. Further works by Satat et al. Bar et al. Alterman et al. all looked into seeing through scattering and Bar et al. offer a simple model for speckle formation. Lastly, the discussion on speckles seem redundant as the paper reduces the model to a simple convolution with a gaussian kernel. This opens the discussion to a whole host of works done on blind and non-blind deconvolution.

The other major issue I have is that all the results and experiments assume a simple convolution model to generate data and then show the recovery based on that model. This means that there is no model mismatch at all. I would like the authors to expand on that.

A minor point: the paper alternates between a differentiable model and a closed form model, which do not overlap. One can have a differential scattering-based model (e.g. Nimier-David, Merlin, et al. "Mitsuba 2: A retargetable forward and inverse renderer." ACM Transactions on Graphics (TOG) 38.6 (2019): 1-17.) that is nonetheless a non-closed form model.

Some relevant work that should be acknowledged and contextualized:
Satat, Guy, Matthew Tancik, and Ramesh Raskar. "Towards photography through realistic fog." 2018 IEEE International Conference on Computational Photography (ICCP). IEEE, 2018.
Antipa, Nick, et al. "DiffuserCam: lensless single-exposure 3D imaging." Optica 5.1 (2018): 1-9.
Berman, Dana, and Shai Avidan. "Non-local image dehazing." Proceedings of the IEEE conference on computer vision and pattern recognition 2016.
Akkaynak, Derya, and Tali Treibitz. "Sea-thru: A method for removing water from underwater images." Proceedings of the IEEE/CVF conference on computer vision and pattern recognition. 2019.
Alterman, Marina, et al. "Imaging with local speckle intensity correlations: theory and practice." ACM Transactions on Graphics (TOG) 40.3 (2021): 1-22.
Bar, Chen, et al. "Single scattering modeling of speckle correlation." 2021 IEEE International Conference on Computational Photography (ICCP). IEEE, 2021.

**Questions:**

See weaknesses.
Overall, I think this is a sound paper. Nevertheless, my concerns are
a) lack of context and comparison with other state-of-the-art approaches that have shown good results in real-world hazy images.
b) I would like the authors to elaborate on the lack of model mismatch by assuming a simple gaussian kernel and then recovering under this assumption. I'm not sure how to evaluate the figure in the appendix. I do not know if other, more physically realistic methods for rendering scattering effects might do much better.

---

> ### Author Response · Authors · 2023-11-17
> **Official Comment by Paper7397 Authors (1/2)**
>
> Thanks for the careful reading and for your detailed feedback. Please see the detailed response below.
>
> > **W1**. The key issue and the reason I chose a rating of 2 for the presentation is the lack of contextualization for this work. While the authors do a good job comparing to some baselines like a TV method and a 2D approach the problem of seeing through scattering media has a long history, which this paper largely ignores. This problem arises in the **(1) context of de-hazing and underwater imaging**.
> I also point out that the inverse problem is very similar to the **(2) DiffuserCam**. **(3) Further works by Satat et al. Bar et al. Alterman et al. all looked into seeing through scattering and Bar et al. offer a simple model for speckle formation**.
> Lastly, the **(4) discussion on speckles seem redundant** as the paper reduces the model to a simple convolution with a gaussian kernel. This opens the discussion to a whole host of works done on blind and non-blind deconvolution.
>
> Please see General Comment 2 & 3.
>
> DiffuserCam has not been mentioned in General Comment 2, so we will provide following response here. We started the derivation by substituting z=0 to derive a Gaussian blur kernel. We emphasize that the distance from the imaging plane from the output plane of scattering media can be non-invasively controlled (e.g. Imaging right in front of the skin). If this **distance becomes large**, it can be considered similar to the case of a DiffuserCam, leading to a speckle PSF rather than a Gaussian psf.
>
> In DiffuserCam, they used this speckle PSF for compressed sensing encoding of 3D information, which has significance. However, due to the Memory effect, the forward model has a significant limitation with a hard limit on the field of view. Therefore, the speckle PSF is unknown without the speckle pattern calibration according to various distances(not applicable to an unseen diffuser), and the object must be spatially incoherent amplitude target. In this point of view, our forward model does not require any calibration to applicate to an unseen diffuser.
>
> > **W2**. The other major issue I have is that all the results and experiments assume a simple convolution model to generate data and then show the recovery based on that model. This means that there is no model mismatch at all. I would like the authors to expand on that.
>
> Please see the General comment 1.
>
> We agree the results on real scattering measurements are required. In the revised manuscript, we add experimental results in real-world optical setting. Our model still works while Poissonian shot noise and camera dark noise were added real scattering measurements.
>
> > **W3**. A minor point: the paper alternates between a differentiable model and a closed form model, which do not overlap. One can have a differential scattering-based model that is nonetheless a non-closed form model.
>
> Thank you for your interesting idea. We will consider it for the future work.
>
> > **W4**. Some relevant work that should be acknowledged and contextualized.
>
> Please see the General comment 3.
>
> In the context outlined above, we undertook a thorough review of all the work in this field, categorizing it into the three specified regimes. Additionally, we engaged in a detailed discussion on the state-of-the-art advancements. Thank you for the suggestion which brings our paper could give the readers a more complete perspective on the field.
>
> > **Q1**. lack of context and comparison with other state-of-the-art approaches that have shown good results in real-world hazy images.
>
> Please see the General comment 2 & 3.
>
> Per your suggestion, we engaged in a detailed discussion on the state-of-the-art advancements including de-hazing and underwater imaging papers suggested by reviewer dFYB. On the other hand, due to the different perspective in the context, utilizing it as a comparative method proves challenging.
>
> > **Q2**. I would like the authors to elaborate on the lack of model mismatch by assuming a simple gaussian kernel and then recovering under this assumption.
>
> Please see General comment 1.
>
> Per your suggestion, we demonstrate a description of the model mismatch in Appendix A.5 of the revised manuscript, especially for noise and phase errors. Our real experiment results serve as a demonstration that our forward model can effectively overcome model mismatch in practical scenarios.

---

> > ### Author Response · Authors · 2023-11-17
> > **Official Comment by Paper7397 Authors (2/2)**
> >
> > > **Q3**. I'm not sure how to evaluate the figure in the appendix.
> >
> > We demonstrate how well our forward model models the imaging of an object located at a distance $d$ behind a dynamic scattering medium.
> >
> > First, according to our forward model, we propagate a phase object by 6/12mm using the angular spectrum method. We take the intensity and convolve it with the Gaussian PSF of the scattering medium, resulting in the image in the simulated column.
> > Then, we position the actual phase target at a distance of 6/12mm from the scattering medium and get the image in the real measurement.
> > The correlation between the two images is calculated using the following formula, with a maximum value of 1:
> >
> > $r(A,B) = \frac{\sum\limits_{m}\sum\limits_{n}(A_{mn}-\bar{A})(B_{mn}-\bar{B})}{\sqrt{(\sum\limits_{m}\sum\limits_{n}(A_{mn}-\bar{A})^2)(\sum\limits_{m}\sum\limits_{n}(B_{mn}-\bar{B})^2)}}$
> > where, $\bar{X}$ is the mean value of image $X$.
> >
> > Per your thorough feedback, we include the description in Appendix A.3 in the revised manuscript.
> >
> > > **Q4**. I do not know if other, more physically realistic methods for rendering scattering effects might do much better.
> >
> > In tasks such as de-hazing or underwater imaging which fall in to the regime 2 of General comment 2, which shares the similar context, the approach suggested by the reviewer may indeed be effective. However, applying the approach to the problem we are dealing with might be challenging.

---

> > > ### Comment · Reviewer_dFYB · 2023-11-20
> > > **Post rebuttal/discussion**
> > >
> > > Thank you for the clarifications. I appreciate you taking the comments and making positive changes to the manuscripts that help with the issue I raised about contextualization. I changed my rating for the presentation to reflect that improvement, though the experimental results are still quite simplistic. Therefore, I kept the contribution rating unchanged. Overall, I think that this rebuttal has swayed me towards a slightly positive rating.

---

> ### Author Response · Authors · 2023-11-22
> **Dear Reviewer dFYB**
>
> Thank you for the extremely helpful suggestions and the positive score. In the light of your comments, we have performed additional experiments featuring more complex scenes (specifically, a Pac-Man video sourced from YouTube). The additional result can be found in Fig. 9 (c) of Appendix A. At the current moment, we could not make significant modifications on our experimental system because of the limited time allowed for open discussion. For the final version after this period, we will further demonstrate our method’s extendibility with additional real-experiment results involving more complex scenes and various scattering configurations.

---

### Official Review · Reviewer_AVri · 2023-10-30

**Soundness:** 3 good
**Presentation:** 3 good
**Contribution:** 4 excellent
**Rating:** 8
**Confidence:** 3

**Summary:**

An original method is proposed to remove dynamic blur in a moving video by taking advantage of spatial and temporal correlations. The  approach consists in introducing temporal aspect in the 2-dimensional posterior sampling (DPS) approach, a similar extension allowing to extend Diffusion models as Video Diffusion Models (VDM). The proposed approach needs tha the diffusion layer is not to thick and that the scene is enlighten with a laser. Comparative experiments are proposed with convincing results.

**Strengths:**

The possibility to take fully advantage of the spatial and temporal correlations is very interesting and useful. The proposed results are very convincing about the superiority of the proposed approach. A source code with example is provided.

**Weaknesses:**

The paper is well introduce and clearly explain but the derivations in appendix are quite hard to follow. It is not derivations but sketchs of derivations. A reference to a technical report with the derivations will be very useful.

**Questions:**

I was not able to find the description about the learning step in the paper. May you tell more about this important step ?

---

> ### Author Response · Authors · 2023-11-17
> **Official Comment by Paper7397 Authors**
>
> Thank you for your encouraging comments. Please see the detailed response below.
>
> > **W1**. The paper is well introduce and clearly explain but the derivations in appendix are quite hard to follow. It is not derivations but sketchs of derivations. A reference to a technical report with the derivations will be very useful.
>
> Per your suggestion, we simplified the explanation on the forward model with technical reports to support the description. The key focus of the forward model lies on coherent propagation and the incoherent scattering kernel. We revised it to emphasize concentration on these aspects. The Gaussian approximation of the scattering kernel shape is a minor point, so we omitted the details.
>
> > **Q1**. I was not able to find the description about the learning step in the paper. May you tell more about this important step?
>
> We have listed the training details in Appendix C for training our diffusion prior. It follows exactly same learning step with VDM which learns target(video data before scattering) distribution which means it does not require any information about physical forward model during learning(training) step. This is one the key advantages of our method that it can handle any variations in forward physical model, once training done.

---

### Official Review · Reviewer_nUZb · 2023-10-31

**Soundness:** 2 fair
**Presentation:** 3 good
**Contribution:** 2 fair
**Rating:** 5
**Confidence:** 5

**Summary:**

This paper proposes applying video diffusion models to the task of reconstructing video captured through scattering media. Specifically, this paper focuses on scenarios where the scattering is approximated by the so-called shower curtain effect, where the forward operator essentially reduces to a Gaussian blur kernel. The proposed method is based on posterior sampling given a pre-trained video diffusion model, supposedly containing prior knowledge of natural videos. The restoration of the original video is equivalent to doing a posterior sampling of the video diffusion model, conditioning on the blurry measurements. The evaluation of the proposed method is mainly done on two existing natural video datasets, and the scattering effect is simulated.

**Strengths:**

- This paper tackles an important problem of imaging through scattering.
- Incorporating diffusion models in the context of imaging through scattering is new.

**Weaknesses:**

- This paper only contains restoration results from **simulated** scattering. No successful restoration on any real-world scattering were demonstrated.
- This paper only focuses on a naive special case of scattering, where the forward operator is trivially a Gaussian blur. Real-world scattering is much more complicated can requires the modeling of phase error caused by the scattering medium.
- Ignoring the significance on the problem of optical scattering, the technical contribution on the algorithm side is very limited. The paper introduces minimal changes to existing approaches that apply the diffusion posterior sampling strategy on other inverse problem tasks.
- This paper does not include literature review on the problem of imaging through optical scattering, and fails to cite recent papers that could give the readers a more complete perspective on the state-of-the-art, such as:
  - Imaging with local speckle intensity correlations: theory and practice, ACM Transactions on Graphics, 2021
  - Guidestar-free image-guided wavefront shaping, Science Advances, 2021
  - Prior-free imaging unknown target through unknown scattering medium, Optics Express, 2022
  - NeuWS: Neural wavefront shaping for guidestar-free imaging through static and dynamic scattering media, Science Advances, 2023.

**Questions:**

What's stopping the proposed method from successfully working on real-world scattering? Can the proposed framework handle more sophisticated forward model?

---

> ### Author Response · Authors · 2023-11-17
> **Official Comment by Paper7397 Authors**
>
> We would like to thank the reviewer for the constructive comments and the thorough feedback. For point-to-point response, see below.
>
> > **W1, Q1**. This paper only contains restoration results from simulated scattering. No successful restoration on any real-world scattering were demonstrated.
>
> Please see the General comment 1.
>
> We agree the results on real scattering measurements are required. In the revised manuscript, we add experimental results in real-world optical setting. Our model still works while Poissonian shot noise and camera dark noise were added real scattering measurements.
>
> > **W2**. This paper only focuses on a naive special case of scattering, where the forward operator is trivially a Gaussian blur. Real-world scattering is much more complicated can requires the modeling of phase error caused by the scattering medium.
>
> Please see the General comment 2.
>
> Our model is the first to combine a wave propagation model and a convolution with a blur kernel in the context of seeing through dynamic scattering media. With these modifications, may deal with the problem of object positioned at arbitrarily distance from scattering medium and the medium’s thickness of far beyond the regime discussed in Regime 3 of general comment 2.
>
> We emphasize that dealing with the problem of object positioned at arbitrarily distance from scattering medium and the medium’s thickness is sufficiently complicated problem representing real-world scattering.
> We believe our model has a great significance in imaging through scatter problem and makes an advancement compared to the previous methods.
>
> > **W3**. Ignoring the significance on the problem of optical scattering, the technical contribution on the algorithm side is very limited. The paper introduces minimal changes to existing approaches that apply the diffusion posterior sampling strategy on other inverse problem tasks.
>
> As reviewer AVri kindly summarize our work, our method shows similar extension allowing to extend Diffusion models as Video Diffusion Models (VDM). The main contribution on the algorithm side is that the exploiting the  spatio-temporal correlation can bring huge improvements not only for generative models but also for inverse problem solvers like DPS. To best of our knowledge, this work is first video reconstruction study using diffusion prior and optical forward model. It is simple but very effective method for real-world, because real-world signals mostly temporally coherent. We firmly believe that our approach will serve as a source of inspiration for both the computer vision and optics research fields.
>
> > **W4**. This paper does not include literature review on the problem of imaging through optical scattering, and fails to cite recent papers that could give the readers a more complete perspective on the state-of-the-art
>
> Please see the General comment 3.
>
> In the context outlined above, we undertook a thorough review of all the work in this field, categorizing it into the three specified regimes. Additionally, we engaged in a detailed discussion on the state-of-the-art advancements. Thank you for the suggestion which brings our paper could give the readers a more complete perspective on the field.
>
> > **Q2**. Can the proposed framework handle more sophisticated forward model?
>
> This method retains all the advantages of DPS. In other words, it can restore measurements even in cases where shot-noise and Gaussian noise are added to the measurement, also for the forward operator is nonlinear.
> The capacity to address shot noise is a pivotal factor, our method highly effective in precisely restoring real optical measurements. In this context, our paper makes a significant contribution by illustrating the feasibility of such a forward model design in practical applications. Consequently, we demonstrate state-of-the-art results within this framework.

---

> > ### Comment · Reviewer_nUZb · 2023-11-20
> >
> > I appreciate the authors' effort in revising the manuscript and updating real-world results. As echoed by Reviewer dFYB, the experimental results are indeed simplistic, and frankly not convincing in the context of imaging through optical scattering. Moreover, the technical contributions on the learning side are also limited (i.e., reapplying DPS). Lastly, the crucial limitation of assuming a naive Gaussian blur as the forward model cannot be overlooked. Therefore, considering its lackluster significance for physical sciences and use of learning representations, I would not rate this paper to be above the threshold for ICLR.

---

> ### Author Response · Authors · 2023-11-22
> **Dear Reviewer nUZb**
>
> Thank you for your invaluable feedback. In the light of your comments, we have conducted additional experiments featuring more complex scenes (specifically, a Pac-Man video sourced from YouTube). The additional results can be found in Fig. 9 (c) of Appendix A.
>
> We fully understand your concern regarding the simplicity of our model. Indeed, the scattering kernel that describes the ensemble-averaged intensity response of a scattering medium (i.e. the intensity response on the output plane for a point source on the input plane) varies depending on the properties of a scattering medium, such as scatterers' shape and size, and refractive indices of scatterers and an embedding medium. Here, based on the assumption that the kernel function smoothly and slowly decays, we simply assumed it as a Gaussian function and proved its effectiveness.
>
> At the current moment, we could not make significant modifications to our experimental system and the proposed forward model because of the limited time allowed for open discussion. For the final version after this period, we will further investigate and demonstrate our method’s extendibility with additional real-experiment results involving more complex scenes with the sophisticated considerations on the types of scattering kernel functions.
>
> Lastly, we reiterate that our work tackles a challenging inverse problem that deals with not only blurred but also absolute-squared, wave-propagated measurements at arbitrary distances from the scattering medium and with variations in medium thickness. All of this achieved using a single trained diffusion prior. We would like to note that the inversion problem relevant to blurring operation and the inversion problem relevant to the absolute square operation of complex wave propagation individually hold their merits and have long been studied in the context of deblurring and holographic imaging. For this challenging combined problem, we demonstrated superior results by fully leveraging the spatial and temporal correlations of diffusion models. We firmly believe that our model holds great significance in this regard.

---

### Author Response · Authors · 2023-11-17
**General Comment by Paper7397 Authors (1/3)**

We thank the detailed comments from the reviewers that helped us in improving our paper. We are encouraged by the positive remarks from the reviewers, “tackles an important problem in optics and incorporating video diffusion models in the problem is new” (nUZb), “shows that using deep learning in video space improves the reconstruction quality” (dFYB), and “possibility to take fully advantage of the spatial and temporal correlations is very interesting and useful and shows convincing and superior results” (AVri).

We have made the following major changes to address the concerns raised from the reviewers.

> **1. Adding experimental results on real-world experiments**

Many of the reviewers’ concerns were pointed at the mismatch between the forward model and real-world measurements. To address this concern, we added real experimental results from real-world optical settings, which we could not finish at the time of original submission. Interesting enough, our model has been shown effective even under the influence of various real-world noise sources, such as Poissonian shot noise (i.e. fluctuation in arriving photon numbers) and camera dark noise. The results and details can be found in Appendix A of the revised manuscript.

> **2. Highlighting the importance of the forward model used in our study**

Reviewer nUZb pointed out that “This paper only focuses on a naive special case of scattering, where the forward operator is trivially a Gaussian blur. Real-world scattering is much more complicated can requires the modeling of phase error caused by the scattering medium” and Reviewer dFYB pointed out that “the discussion on speckles seem redundant as the paper reduces to a simple convolution with a Gaussian kernel”.

We respectfully disagree with the Reviewer’s comments. There have been many previous efforts to solve scattering problems based on computational algorithms or physical corrections. We may categorize those previous approaches based on their working thickness and scattering strength – 1) thick and strongly scattering regime, 2) thick and weakly scattering regime, and 3) thin regime. Here, we carefully described the previous approaches with categorizing them into three categories. Subsequently, we will highlight the importance of the forward model used in our study.

**Regime 1**:

Wavefront shaping approaches fall into the regime 1 – thick and strongly scattering media. As demonstrated in previous iterative optimization techniques, transmission matrix measurements, and optical phase conjugation, most wavefront shaping approaches aim to physically control the input wavefront based on spatial light modulators with many degrees of freedom, typically more than $10^3$, and thereby, to deterministically manipulate the output optical fields based on the both spatial and temporal coherence of optical fields (i.e. interference effect). As long as the medium is static during this deterministic control of wavefront, we may reverse some effect of optical scattering without any restrictions on the medium’s thickness and scattering strength. The following references fall into this category.
* Controlling waves in space and time for imaging and focusing in complex media, Nature photonics, 2012 - reviews
* Focusing coherent light through opaque strongly scattering media, Optics Letters, 2007 – iterative optimization
* Measuring the Transmission Matrix in Optics: An Approach to the Study and Control of Light Propagation in Disordered Media, Phys. Rev. Lett., 2010 – transmission matrix
* Optical phase conjugation for turbidity suppression in biological samples, Nature photonics, 2008 – phase conjugation

**Regime 2**:

Dehazing and underwater imaging problem falls in to the regime 2 – thick and weakly scattering media. More specifically, setting $L$ and $l_s$ as the thickness and the mean scattering path of the turbid medium, $L$ is smaller than or comparable to around 10 × $l_s$ in this regime. Based on the Beer-Lambert law, the ballistic component of light (i.e. image-carrying signal) decays exponentially with the factor of exp($-L/l_s$) while other components (i.e. scattering background noises) are back-scattered or forward-scattered. Most approaches in the regime 2 aim to computationally remove the scattering background noises and compensate for the path-dependent exponential factor for image-carrying signals. The references suggested from dFYB fall into this regime.
* Towards photography through realistic fog, ICCP, 2018
* Non-local image dehazing, CVPR, 2016
* Sea-thru: A method for removing water from underwater images, CVPR, 2019

---

> ### Author Response · Authors · 2023-11-17
> **General Comment by Paper7397 Authors (2/3)**
>
> **Regime 3**:
>
> The computational approaches based on ‘memory effect’ and ‘Shower-curtain effect’ fall in to the regime 3. Our proposed forward model is most relevant to this regime. However, there is clear distinction from the previous approaches from the models used in the previous studies.
>
> Existing studies related to the ‘Memory effect’ and ‘Shower curtain effect’ have modeled scattering as an **infinitesimally thin phase mask. With this assumption, the short-range correlation is retained for arbitrarily large input angle [1]**. Therefore, setting the object/detection planes far apart from a scattering medium, the autocorrelation function of object pattern becomes identical to the autocorrelation function of the measured speckle pattern. Based on Wiener–Khinchin theorem, the equivalence of the two autocorrelation functions is translated to the equivalence of the power spectral densities of two patterns (i.e. Fourier magnitude) so that the phase retrieval algorithms may be applied to recover the object pattern.
>
> **It should be noted that this assumption breaks down in the real world, limiting the achievable field of view in the ‘memory effect’ scheme (where the camera is positioned far apart from a scattering medium) and the achievable resolution in the ‘shower curtain effect’ scheme (where the camera is conjugated to the output plane of a scattering medium)**.
>
> We also note that two wavefront shaping approaches suggested from uNZb [2], [3] are also treating scattering media as the infinitesimally thin phase mask. So, it corrects only the single-layer phase distortion (i.e. phase delay map of thin scattering media) and is only applicable when the dynamic nature of scattering media is slow enough to be followed by the spatial light modulator with the refresh rate of 60 - 1,000 Hz.  Again, such approach would not work when the scattering medium becomes thick so that the response of scattering medium cannot be described as a position wise multiplication of input field and the phase map [4].
>
> Only recently, there have been a few attempts to take account of imaging configurations and the effect of finite thickness of scattering media based on thorough considerations on scattering models and the range of correlation effect. Notably, in [5], [6] suggested by reviewer uNZb and dFYB, the restriction on the field of view due to the finite range of memory effect is relaxed with the local support to enhance the effect of local correlation.
>
> Our model is the first to combine a wave propagation model and a convolution with a blur kernel in the context of seeing through dynamic scattering media. Interestingly, in the problem of seeing through dynamic scattering media, light propagation from an object to a scattering medium is coherent process, which can be only described on the basis of complex amplitude field, but through dynamic scattering, the wave lose its spatial and temporal correlation, which may be properly described on the basis of intensity-based scattering kernel, which in our study is approximated as a Gaussian kernel.
>
> **With these modifications, in contrast to the previous approaches in the Regime 3, one may deal with the problem of object positioned at arbitrarily distance from scattering medium and the medium’s thickness of far beyond the regime discussed in Regime 3 (i.e. thin phase mask or medium that can be treated as single scattering regime)**. But, the hybrid model is computationally more difficult. We handle with the additional use of temporal correlation, which has been largely ignored in previous studies but proven powerful as presented in our study. In this aspect, we believe our model has a great significance in the problem of imaging through scatter and makes an advancement compared to the previous approaches in Regime 1, 2, and 3.
>
> [1] Memory Effects in Propagation of Optical Waves through Disordered Media, Phys. Rev. Lett., 1988
>
> [2] Guidestar-free image-guided wavefront shaping, Science Advances, 2021
>
> [3] NeuWS: Neural wavefront shaping for guidestar-free imaging through static and dynamic scattering media, Science Advances, 2023
>
> [4] Measuring the Transmission Matrix in Optics: An Approach to the Study and Control of Light Propagation in Disordered Media, Phys. Rev. Lett., 2010
>
> [5] Imaging with local speckle intensity correlations: theory and practice, ACM TOG, 2021
>
> [6] Single scattering modeling of speckle correlation, ICCP, 2021

---

> > ### Author Response · Authors · 2023-11-17
> > **General Comment by Paper7397 Authors (3/3)**
> >
> > > **3. Strengthen the contextualization with additional literature reviews on the problem**
> >
> > We thank the reviewer for pointing out this important omission. In the original manuscript, we have focused on delivering the key concept of utilizing spatiotemporal correlation. Based on the suggested citation from Reviewer uNZb and dFYB, we supplement the contextualization of the scattering of our interest. As thoroughly discussed above, we classified the scattering problems in three regimes depending on the medium’s thickness and scattering strength and introduced key concepts and models developed in previous studies. With this effort, we hope that the readers would have a more comprehensive perspective on the problem of seeing through dynamic scattering media and the proposed forward model in relation to previous studies. The modified details can be found in Section 2(Related works) of the revised manuscript.
> >
> > More details and explanation on the questions are elaborated in the following responses.

---

### Comment · Reviewer_AVri · 2023-11-20
**Thanks**

Thank you for the very constructive responses.

---

> ### Author Response · Authors · 2023-11-22
> **Dear Reviewer AVri**
>
> Thank you for the discussion and the positive score.

---

### Meta-Review · Area_Chair_HNZv · 2023-12-06

**Metareview:**

This paper proposes the use of diffusion models for reconstructing video data through scattering media.

**Justification For Why Not Higher Score:**

The reviewers agree that the novelty of the proposed method is limited and below ICLR expectations.

**Justification For Why Not Lower Score:**

N/A

---

### Decision · Program_Chairs · 2024-01-16

Reject